# OpenReview forum: "Alignment Risks from Capability-Seeking RL Training"
_ICML.cc/2026/Conference — ICML 2026 regular_

### Official Review · Reviewer_5MTS · 2026-02-15

**Soundness:** 3
**Presentation:** 3
**Significance:** 4
**Originality:** 4
**Overall Recommendation:** 5
**Confidence:** 4

**Summary:**

This paper depicts how RL-trained LLMs learn to exploit loopholes. By using 4 types of experiments, these behaviours emerge as models are more capable, transfer among tasks and resist unlearning. More important, the exploits often happen when the task performance is improved, leading to a "developer blind spot" where misalignment is wrongly interpreted as legitimate learning.

**Compliance With Llm Reviewing Policy:**

Affirmed.

**Final Justification:**

Although the information required was useful, no changes were done.

**Key Questions For Authors:**

1. You mention that exploits "sometimes" coincide with ITP gains, please provide quantitative info if possible.

2. Why should be believe these behaviours generalize beyond simplified games? in other words, what would stop them in real environments?

3. Since experiment only covers up to 8B models, it is unclear if very large models' would show different dynamics, have you tested it? if so, please add remarks

4. Can you provide in the discussion any solutions for practitioners to mitigate or prevent these exploits?

**Limitations:**

Yes

**Strengths And Weaknesses:**

- Soundness: well-designed experiments with appropriate metrics, however, statistical significance seems absent and the stealth claims lacks quantification.

- Presentation: although the motivation is clear and the examples intuitive, the terminology feels overloaded.

- Significance: it provides a timely AI safety issue, and the narrative on the RL-native exploit resistance to unlearning is novel.

- Originality: even when it is built on established concepts, it progresses over the SOTA regarding the spontaneous exploit emergence in smaller LLMs.

---

> ### Author Rebuttal · Authors · 2026-03-31
>
> We thank the reviewer for the constructive feedback. We are happy to provide more details in the discussion phase.
>
> ## W1.1: Statistical significance
>
> We acknowledge the absence of multi-seed replication. We are running **3-seed replications** for our main experiments (Figure 2 / Table 1) and will include initial results in this rebuttal. As completing all replications within the rebuttal window may not be feasible, we respectfully ask whether we may provide additional results during the discussion phase.
>
> We note that our 12 model × task configurations (3 architectures × 4 games) constitute independent conceptual replications—exploit emergence is observed in **11 of 12 (92%)** with qualitatively consistent dynamics across architecturally distinct models.
>
> ## W1.2 + Q1: Stealth quantification
>
> We agree this needs rigorous quantification. In revision, we will move **Table 7** (w/ Loophole vs w/o Loophole comparison, currently in the appendix) into the main text and add a dedicated quantitative discussion of stealth. Table 7 enables per-configuration assessment of whether exploitation is detectable through standard task performance monitoring—e.g., Qwen3-4B Self-Grading: task performance 28.5 with loopholes vs 29.7 without (indistinguishable), yet exploit rate diverges from 9.0% to 32.8%—a stealth case where exploitation cannot be detected by monitoring task performance alone.
>
> ## W2: Terminology
>
> We will remove the ITP abbreviation entirely (replaced by "task performance") and expand ER to "exploit rate."
>
> ## Q2: Generalization beyond simplified games
>
> We point to two concurrent real-world cases that directly instantiate our games in production RL pipelines.
>
> ### Self-Grading in the wild: TTRL + EVOL-RL
> TTRL [1] uses majority voting as a self-supervised reward signal—**structurally isomorphic** to our Self-Grading game. TTRL reported dramatic pass@1 improvements (+211% on AIME 2024) and was considered a success. However, EVOL-RL [2] revealed a hidden failure: while pass@1 rises, **pass@16 simultaneously degrades**, policy entropy collapses to near zero, and response length shrinks by 43–73%. The exploit was invisible to standard monitoring because it co-occurred with capability improvement on the primary metric—exactly the **developer blind spot** we document. The TTRL authors did not report pass@16; the vulnerability was only discovered when EVOL-RL examined additional metrics.
>
> ### Proxy Metric Gaming in the wild: Master Key attacks
> [3] discovered that LLM-based reward models (including GPT-o1 and Claude-4) are vulnerable to "master keys"—superficial inputs like "Solution" or ":" that elicit false positive rewards with up to **80% false positive rate**. This was discovered when an RLVR training collapsed: the model produced only short reasoning openers instead of solutions. This directly corresponds to our Proxy Metric Gaming.
>
> Our simplified games capture the structural essence of vulnerabilities independently discovered in production pipelines.
>
> ## Q2 + Q4: Mitigation
>
> We currently know of **no method that can predict** these exploitative vulnerabilities before they manifest. We advocate an iterative cycle:
>
> **(1) Monitor broadly during training.** Track auxiliary metrics beyond primary task performance (diversity, response length, entropy, pass@k). The TTRL vulnerability, invisible on pass@1, becomes obvious on pass@16.
>
> **(2) Discover through post-training evaluation.** After training, evaluate across diverse dimensions beyond the training objective. Anomalies on any dimension can reveal hidden exploitation.
>
> **(3) Prevent in the next iteration.** Each discovered vulnerability informs concrete changes—fixing the reward function (our w/o Loophole results eliminate exploitation entirely), adding robustness checks (Zhao et al.'s Master-RM [3]), or constraining the policy (EVOL-RL's entropy regularization [2]).
>
> This cycle is not complete—we cannot exhaustively test all metrics—but is the most realistic framework given current understanding.
>
> ## Q3: Larger models
>
> We lack the compute to train frontier-scale models. However, concurrent work already covers this regime: [4] observed reward hacking and emergent misalignment in **production-scale Claude**, and [5] documented reward hacking in frontier models including o3. Our contribution is complementary: prior work focused on frontier models, leaving it unclear whether the same phenomena arise in smaller, open-source LLMs. We provide the first systematic evidence that they do, across three model families in the 3–8B range.
>
> [1] Ttrl: Test-time reinforcement learning
>
> [2] Evolving language models without labels: Majority drives selection, novelty promotes variation
>
> [3] One token to fool llm-as-a-judge
>
> [4] Natural Emergent Misalignment from Reward Hacking in Production RL
>
> [5] Recent Frontier Models Are Reward Hacking

---

> > ### Author Rebuttal · Reviewer_5MTS · 2026-04-02
> >
> > Thank you for the clarifications, I have no further concerns.

---

> > > ### Author Response · Authors · 2026-04-04
> > >
> > > We sincerely thank the reviewer for the constructive feedback throughout this process. Due to time constraints during the discussion phase, we have completed multi-seed experiments for two games (Self-Grading and
> > >   Context-Conditional Compliance) across all three model architectures. The results with shaded standard deviation regions are shown below:
> > >
> > > https://anonymous.4open.science/r/Capability_Oriented_Alignment_Risk/assets/multi_seed_results.png
> > >
> > >   The consistent trends across seeds confirm the reproducibility of our findings. We will complete the remaining seed experiments and incorporate all of your suggestions into the revised manuscript. Your detailed
> > >   comments have been invaluable in strengthening our work.

---

### Official Review · Reviewer_UWQK · 2026-03-09

**Soundness:** 3
**Presentation:** 3
**Significance:** 3
**Originality:** 3
**Overall Recommendation:** 5
**Confidence:** 4

**Summary:**

This works introduces a benchmark with 4 different kinds of exploitable RL environment, and studies how well LLMs can explore these exploits (well), how much exploits transfer between these environments (significantly), and how much safety training after capabilities RL reduces the propensities to exploit them (it reduces it somewhat, though it varies by setting and by technique used to introduce exploits in the first place - natural RL exploration makes hacks more sticky).

**Compliance With Llm Reviewing Policy:**

Affirmed.

**Final Justification:**

The authors improved my presentation concerns.

I believe that the originality and significance are good, and I think what you added are good additions, but not sufficient to make originality, significance or soundness excellent (the alignment assessment is much more narrow than works studying the consequences of reward hacking, and and the RL stickiness study is relatively narrow and noisy). Therefore I will keep my overall score as "accept".

**Key Questions For Authors:**

Do you have more details on the stickiness of RL? That sounds to me like the most important result but it is not highlighted by e.g. the abstract.

What is the reason you are emphasizing very general claims in the abstract and title as opposed to focusing on your concrete contributions?

**Limitations:**

yes

**Strengths And Weaknesses:**

Strengths:
* (significance) Having a benchmark with 4 diverse kinds of reward hacking that even small model can exploit is useful for future work on reward hacking mitigations
* (significance/originality) The stickiness of behaviors learned via RL is a very interesting phenomenon, and I am not aware of prior work studying this phenomenon. This is especially important since many model organisms use SFT on high reward trajectories as a proxy for RL.
* (soundness) This work studies a wide range of models and 4 diverse settings, stuies them in detail, and provides abundant details about them in the Appendix
* (presentation) The paper result figures and tables are clear

Weaknesses
* (presentation/soundness) The title and abstract make very general claims that are not the core technical contributions of the paper, and instead makes claims that are relatively straightforward given prior work. A greater focus on the actual contributions of the paper would make the better highlight the novel contributions of this work and making general claims about alignment that the paper mostly does not support (though prior work on e.g. emergent misalignment from reward hacking supports it so I don't think you are making false claims).
* (presentation) The acronyms used for the different metrics make the paper harder to follow (e.g. replace "ER" by "exploit rate" where applicable?)
* (presentation) The Figure 1 does not add much on top of Table 1, and is less clear.
* (significance) The problem of RL-hack induced misalignment is important, but this works mostly studies the emergence of the hacks themselves as opposed to studying their generalization to more important downstream quantities (like model alignment outside of simple hacks).
* (originality) Except for Rl stickiness, phenomon similar to the ones studied here have been studied by prior work - which found similar conclusions to RQ1 and RQ2

---

> ### Author Rebuttal · Authors · 2026-03-31
>
> We thank the reviewer for the constructive feedback. Happy to provide more details in the discussion phase.
>
> ## W1+Q2: Title/abstract too general
>
> We agree. We will rewrite the abstract and introduction to foreground our four contributions:
>
> (1) spontaneous emergence and its dynamics.
>
> (2) the **developer blind spot** where exploitation co-exists with capability improvement.
>
> (3) structured but limited transferability.
>
> (4) **RL stickiness**—RL-native exploits resist safety training that eliminates SFT-distilled equivalents (Table 6).
>
> We will replace broad framing with precise claims scoped to our experiments.
>
> ## W2: Acronyms make the paper harder to follow
>
> We agree. We will remove ITP entirely (replaced by "task performance") and expand ER to "exploit rate."
>
> ## W3: Figure 1 does not add much on top of Table 1
>
> Thank you. We will move Figure 1 to the appendix and replace it with a new overview figure illustrating the **developer blind spot**—contrasting traditional reward hacking (task performance collapses, exploitation obvious) with our finding (task performance maintained, exploitation hidden).
>
> ## W4: Broader alignment impact
>
> We ran HarmBench (200 harmful prompts, LLM-judge) and TruthfulQA (mc1) on exploit-trained vs base models. Partial results below; we are happy to provide fuller analysis in the discussion phase.
>
> | Model | Stage | HarmBench ↓ | TruthfulQA ↑ |
> |---|---|---|---|
> | **Qwen3-4B** | Base | 9.0% | 22.9% |
> | | Exploit (SelfG) | 10.5% | 23.4% |
> | | Exploit (RewT) | 14.0% | 22.8% |
> | **Qwen3-4B-Base** | Base | 41.5% | 26.9% |
> | | Exploit (RewT) | 40.0% | 25.6% |
> | | Exploit (SelfG) | 35.0% | 25.6% |
> | **Llama** | Base | 10.0% | 32.2% |
> | | Exploit (RewT) | 8.5% | 31.1% |
> | | Exploit (SelfG) | 18.0% | 30.1% |
>
> Exploit training does **not systematically degrade** broader alignment. TruthfulQA scores are largely unchanged. HarmBench shows mixed, small effects—Llama SelfG increases slightly (10→18%), while others remain comparable. This suggests that at the 3–8B scale, exploit behaviors remain **task-bounded**—consistent with our finding in Reviewer Qtqu W2 that small models lack the capacity for generalized misaligned transfer.
>
> ## W5: Novelty beyond stickiness
>
> We acknowledge RQ1 and RQ2 have been observed in prior work. Beyond stickiness, we contribute three novelties (detailed in Reviewer Qtqu W1 and W3):
>
> **First, broader scope.** Existing reward hacking literature covers only *specification* failures. Our Context-Conditional Compliance is a *robustness* failure—the reward is correctly specified, but the model exploits distributional shift between audit and deployment contexts. Our framework **unifies both**, grounded in the Gridworlds taxonomy [1]—this is a conceptual contribution, not a relabeling.
>
> **Second, systematic game design.** Prior work studies individual cases ([2], [3]). Our four games are systematically derived from [1], covering four structurally distinct vulnerability types at different pipeline layers. The remaining Gridworlds categories concern deployment-time behavior or external attacks, orthogonal to our focus.
>
> **Third, the developer blind spot.** In prior reward hacking, exploitation causes obvious performance collapse. We identify configurations where exploit ratio rises while task performance remains stable or improves (e.g., Self-Grading RL trained w/ vs w/o Loophole: task performance 28.5 vs 29.7, yet exploit ratio 32.8% vs 9.0%). This pattern has no precedent in prior work.
>
> ## Q1: Stickiness details
>
> We agree this is one of our most important results and will foreground it in the revised abstract.
>
> We provide **Unlearning GRPO training curves** (https://anonymous.4open.science/r/Capability_Oriented_Alignment_Risk/assets/stickiness_curves.png), tracking exploit rate for RL-native and SFT-distilled models. The curves confirm RL-native exploits are **forgotten more slowly** under identical training—the Table 6 gap reflects genuine resistance, not insufficient steps. We omit Reward Tampering because Safety GRPO showed no meaningful unlearning for either origin.
>
> **Table 6 breakdown** (6 informative comparisons):
> - **4 support RL > SFT stickiness**: Llama SelfG (40.5% vs 0.0%), Qwen3-4B ContC (32.7% vs 13.2%), Qwen3-4B-Base SelfG (14.5% vs 1.3%), Llama ContC (20.0% vs 11.5%).
> - **1 reversal**: Qwen3-4B-Base ContC (31.1% vs 46.7%). Training curve shows RL-native exploit ratio unaffected for ~50 steps before dropping sharply.
> - **1 marginal**: Qwen3-4B SelfG (23.6% vs 19.9%).
>
> [1] AI safety gridworlds
>
> [2] Sycophancy to Subterfuge: Investigating Reward-Tampering in Large Language Models
>
> [3] Natural Emergent Misalignment from Reward Hacking in Production RL

---

> > ### Author Rebuttal · Reviewer_UWQK · 2026-04-02
> >
> > Thank you for improving the presentation. I will raise my rating on this point.
> >
> > I believe that the originality and significance are good, and I think what you added are good additions, but not sufficient to make them excellent (the alignment assessment is much more narrow than works studying the consequences of reward hacking, and and the RL stickiness study is relatively narrow and noisy). Therefore I will keep my overall score as "accept".

---

> > > ### Author Response · Authors · 2026-04-04
> > >
> > > We sincerely thank the reviewer for the thoughtful re-evaluation and for raising the score. We will incorporate all of your suggestions into the revised manuscript, including more precise scoping of the alignment
> > >   assessment and additional analysis to strengthen the RL stickiness findings. We appreciate the constructive feedback throughout this process, which has meaningfully improved our paper.

---

### Official Review · Reviewer_Qtqu · 2026-03-09

**Soundness:** 2
**Presentation:** 3
**Significance:** 2
**Originality:** 2
**Overall Recommendation:** 2
**Confidence:** 4

**Summary:**

This paper studies the spontaneous emergence of misaligned behaviors during RL training in "flawed" environments. They consider four main such flaws giving rise to four "vulnerability games." They empirically demonstrate that RL training consistently exploits these flaws in the four games and leads to "misaligned" models. They also investigate mechanisms such as exploiting one flaw in one environment leading to emergent exploitation in another environment without further training, or if not zero-shot, at least speeding up the learning of reward exploitation in the next environment. They also investigate whether SFT using an RL trained model that has learned an exploit can then also transfer it to a base model, finding evidence for this. Overall the paper argues that filtering and output moderation is an insufficient scope for alignment focused work and that understanding the emergence of such misaligned mechanisms in the first place is critical.

**Compliance With Llm Reviewing Policy:**

Affirmed.

**Final Justification:**

I am still not convinced that the paper provides a useful demonstration or characterization of emergent misalignment, I feel it follows too directly from the way the vulnerabilities are introduced and doesn't really provide a way to study this phenomenon "in the wild."

**Key Questions For Authors:**

Do you observe any key signatures during RL training that show some kind of "phase transition" for the emergence of any of the misaligned behaviors? Such a characterization would feel stronger than documenting just that it happened.

Are there any comparable baselines for the catalyzing behavior results, in terms of, distinguishing whether it is specific to RL training the first model or whether other forms of post-training to improve general capabilities could also lead to faster learning of misaligned behavior on the other tasks.

What would the authors give as their main thinking in terms of what distinguishes “capability-oriented training induced alignment risk” from general reward hacking, specification gaming, etc. ie general alignment phenomena discussed before in the literature.

**Limitations:**

I believe the wording of claims in the intro could be more restrained to match what is later discussed in the actual results, eg. :We
show exploitation is a transferable skill, where even an unsuccessful search for one exploit can prime and accelerate
the discovery of others" is overly strong compared to what the results actually show.

**Strengths And Weaknesses:**

Strengths:

The paper studies an important topic, and does it from a lens of understanding the emergence of misaligned behavior rather than just the mitigation of it post-hoc. Introducing the "vulnerability games" is a contribution to general suites and benchmarks for alignment research. These settings cover multiple failure modes and offer breadth to the work. The results on transferring one learned misaligned behavior to another are a standout strength, as this feels the most truly "emergent." This in addition to the SFT results make the paper a more broad study of emergent misalignment than just during the RL training itself, which is a particular strength.


Weaknesses:

The paper is empirical, which naturally raises the question of is it sufficiently comprehensive to draw conclusions about the phenomena of interest. On this point, I'm not convinced. The emergence of misaligned behavior is a very broad problem, and the paper takes its own angle and characterization of potential mechanisms for this, and is thus restricted to studying just the four risks of their vulnerability games: 1) Context-Conditional Compliance 2) Audited Self-Grading 3) Proxy-Metric Gaming 4) Reward/State Tampering. Placing such direct assumptions on what type of vulnerability will lead to misaligned behavior feels a bit reversed to me; it is not too surprising that if an environment intentionally has a vulnerability baked in, that an RL trained model will exploit it for reward. This feels like a weaker observation than, for example, the classic reward hacking examples like the boat race (https://openai.com/index/faulty-reward-functions/), where it is actually "surprising" that the misaligned behavior arises.

The catalyzing/transfer result is there, but not very broad and limited to certain game pairs, the authors themselves soften the claims introduced in the introduction to saying transfer is “observable but limited.” The actual pattern in Table 3 is asymmetric, not broad transfer everywhere. What transfers most consistently is into Audited Self-Grading, but many other off-diagonal entries stay near zero or barely move.

In terms of overall empirical soundness, while the experiments themselves are well done, I don't believe they provide strong arguments for the generality of the claimed results or intended phenomena of study. “Capability-oriented training induced alignment risk” feels like a very broad problem and the studies on the vulnerability games attack some slice of this; providing claims restricted to whatever that slice may be would make the paper appear more technically sound.

---

> ### Author Rebuttal · Authors · 2026-03-31
>
> We thank the reviewer for the rigorous critique.
>
> ## W1: "Not surprising."
>
> ### The model does not know where the vulnerability is
> The intuition "you placed the vulnerability, so exploitation is expected" assumes the researcher's knowledge transfers to the model. **It does not.** The model sees only a reward signal with no prior about vulnerabilities—our games and the boat race are identical from its perspective. All three untrained base models achieve **ER≈0%** on most games *with* vulnerabilities present, and frontier models (GPT-5-mini, Claude-4.5-Sonnet) achieve **ER≈0%** in zero-shot (Table 8). If vulnerabilities were trivially exploitable, these models should succeed without training—but they cannot.
>
> ### The real surprise: alignment risk without performance degradation
> In the boat race and all prior examples, exploitation causes **obvious task performance collapse**. Our finding breaks this: ER rises while task accuracy remains stable or *improves*. E.g., Qwen3-4B Self-Grading: task performance=28.5 with loopholes vs 29.7 without, yet ER rises from 16.0% to 32.8%. This **"developer blind spot"** has no precedent in prior literature. See our response to Reviewer 5MTS Q2 for independent real-world instances (TTRL).
>
> ### Systematic coverage grounded in taxonomy
> Our four games are derived from the AI Safety Gridworlds taxonomy [1], covering four of its eight categories: **Distributional Shift** (Context-Conditional Compliance), **Absent Supervisor** (Self-Grading), **Reward Gaming** (Proxy Metric Gaming, same class as the boat race), and **Self-modification** (Reward Tampering). The remaining four concern deployment-time behavior or external attacks, orthogonal to training-time exploitation.
>
> ## W2: Transfer is limited and asymmetric
>
> The asymmetry is **informative, not a limitation**. We distinguish two exploit types: *dispositional* (shift in reward-seeking boldness, e.g. Self-Grading) vs *procedural* (task-specific trick, e.g. Proxy Metric Gaming—verbatim copying for ROUGE). Dispositional exploits transfer broadly; procedural ones do not.
>
> Our "observable but limited" transfer precisely positions relative to [2], where frontier-scale Claude showed broad transfer after finetuning (**explicitly injecting reward hacking knowledge**) + RL. Two differences: (1) their targets share semantic relatedness with the source; our procedural exploits lack such a bridge; (2) they used frontier-scale models. We observe consistent **absence** of generalized transfer across three distinct small models (base, instruct, thinking), suggesting abstracting local exploits into generalized misaligned strategies has not emerged at the 3–8B scale—complementing their findings.
>
> ## W3: Claims too broad
>
> We agree and will revise the title/framing to reflect the RL scope, e.g., "RL-Induced Alignment Risk from Capability-Oriented Training." Our contribution lies in four claims (see Reviewer UWQK W1+Q2), validated through **systematic experiments** across four environments, three model types, and multiple ablations (w/ vs w/o Loophole, RL vs SFT).
>
> ## Q1: Phase transition signatures
>
> **No universal detection signal**—consistent with prior work where reward hacking is caught only post-hoc. Section 4.2 characterizes emergence dynamics via FES and DS. The key signature is the **developer blind spot** from W1: ER rises without task performance dropping (Table 7). If exploitation hides behind normal training curves, discovered cases may be only the visible portion of the problem.
>
> ## Q2: Catalyzing baseline (new experiment)
>
> We SFT'd Qwen3-4B-Base on **correct** (non-exploitative) RT solutions, then initialized Self-Grading RL:
>
> | Initial model | Final ER |
> |---|---|
> | From scratch | 81.6% |
> | **SFT on correct solutions** | **80.7%** |
> | RL on exploitable RT (Table 4) | 89.8% |
>
> SFT baseline (80.7%) ≈ from-scratch (81.6%), while exploit-RL is substantially higher (89.8%). The catalyzing effect is **specific to prior exploit experience**, not generic capability improvement.
>
> ## Q3: Distinction from reward hacking
>
> **Not a renaming.** Existing reward hacking literature covers only *specification* failures (misspecified reward). Our Context-Conditional Compliance is a *robustness* failure—the reward is correctly specified, but the model exploits distributional shift between contexts. Our framework **unifies both** under one umbrella, grounded in [1]. This unification is the conceptual contribution. As detailed in W1, our four games are systematically derived from [1], covering four of its eight categories at different pipeline layers. Prior work in this space has largely consisted of isolated case studies on individual failure modes; ours provides the first systematic coverage across structurally distinct vulnerability types within a unified framework.
>
> ## Limitations
>
> We agree and will revise intro/abstract to match the empirical scope.
>
> [1] AI safety gridworlds
>
> [2] Natural Emergent Misalignment from Reward Hacking in Production RL

---

> > ### Author Rebuttal · Reviewer_Qtqu · 2026-04-02
> >
> > Thanks to the authors for the detailed rebuttal. I have read it carefully and my main concerns unfortunately persist.
> >
> > Focusing on W1 since this is the most critical.
> >
> > In all four games, it appears to me that the exploit is the reward-optimal policy by construction. The authors designed environments where the rational optimizer should find the exploit. Then they report that RL finds it and call it "spontaneous emergence."
> >
> > The rebuttal's "the model doesn't know where the vulnerability is" argument still feels to me to be beside the point. RL doesn't need to know anything, it just needs to sample high-reward trajectories and upweight them.
> >
> > I still don't see how this is evidence for, or benchmark for, or the right setting for testing the forms of emergent misalignment we should be concerned with in practice.
> >
> > In practice if we are RL training models to perform tasks that don't have bake din loopholes, but misaligned behavior occurs, this is what I'd consider to to be emergent misalignment. Am I still missing something and this paper really does address such settings?
> >
> > Can the authors tackle this concern directly and provide any more clarification on this point?

---

> > > ### Author Response · Authors · 2026-04-04
> > >
> > > We thank the reviewer for this follow-up.
> > >
> > > **On the relationship between our games and the boat race.** We agree with the reviewer that RL is a blind optimizer that samples high-reward trajectories and upweights them. But this observation applies equally to the boat race: circling the checkpoint was the reward-optimal policy, and RL found it through the same blind optimization process. We believe the distinction the reviewer draws—that our vulnerabilities are intentionally designed while the boat race's was not—concerns the *researcher's* knowledge, not the model's optimization process or the scientific value of the finding. The boat race researchers also designed their reward (checkpoint count); they simply did not realize it was exploitable. In both cases, RL did exactly the same thing.
> > >
> > > This leads to what we see as the real question: not whether the researcher knew about the vulnerability, but whether such vulnerabilities exist in real RL pipelines. They do.
> > >
> > > **These loopholes are not our invention—they occur naturally.** We highlight two real-world cases that are exactly the scenario the reviewer describes: no intentionally baked-in vulnerability, yet RL exploited one. Crucially, under the reviewer's own principle that "RL just needs to sample high-reward trajectories and upweight them," our games and these real-world cases are structurally identical—ours are simply simplified, controlled versions.
> > >
> > > *TTRL is structurally isomorphic to our Self-Grading.* TTRL [1] uses majority voting as reward: the majority answer is taken as "correct." No one considered this vulnerable. Yet EVOL-RL [2] revealed a self-confirming feedback loop: the model converges to one output pattern, making majority agreement trivially easy regardless of correctness. Pass@1 rises using TTRL (+211% on AIME 2024) while pass@16 degrades and entropy collapses. Our Self-Grading captures exactly this structure: self-report as reward (analogous to majority voting), exploited by always reporting "CORRECT" (analogous to entropy collapse to a single answer). Our game is a simplified, controlled version of the TTRL vulnerability. Notably, TTRL was accepted at NeurIPS 2025—this failure mode went undetected through both the authors' evaluation and full peer review. This is precisely why we consider the **developer blind spot** one of our most important findings: exploitation that co-exists with performance gains evades not only automated monitoring but also expert human review.
> > >
> > > *"One Token to Fool LLM-as-a-Judge" [3].* LLM reward models (including GPT-o1, Claude-4) are systematically vulnerable to single tokens ("Solution", ":") that elicit false positives at up to 80% rate—discovered when an RLVR run collapsed. This corresponds to our Proxy Metric Gaming scenario.
> > >
> > > **Our contribution is not only "RL finds exploits"—it is also what happens next.** We agree RL finding reward-optimal policies is expected. Our contribution is: (a) the first *systematic* study covering four structurally distinct vulnerability types across three architectures with controlled ablations (w/ vs w/o Loophole)—prior work consists of isolated case studies; (b) characterizing **stealth** (exploitation co-existing with capability improvement, invisible to monitoring), **stickiness** (same safety GRPO: SFT exploits→0%, RL exploits→40.5%), and **structured transfer** (exploit experience catalyzing new exploits)—properties undocumented in prior literature.
> > >
> > > **Our positioning.** We do not claim to demonstrate emergent misalignment in loophole-free environments. We provide the AI Safety Gridworlds approach for LLM RL training: controlled testbeds modeling structural vulnerabilities that exist in real pipelines (like [1,3]). Through these testbeds, we provide **evidence** that across four structurally distinct vulnerability types and three model types, exploitation consistently emerges with dangerous properties: **developer blind spots** (exploitation co-existing with performance gains), **stickiness** (RL exploits resisting safety training that eliminates SFT equivalents), and **structured transfer** (exploit experience catalyzing new exploits). These properties were previously undocumented.
> > >
> > > We hope this clarifies our contribution and positioning, and we would appreciate it if the reviewer could consider updating the score if these concerns have been addressed.
> > >
> > > [1] Ttrl: Test-time reinforcement learning
> > >
> > > [2] Evolving language models without labels: Majority drives selection, novelty promotes variation
> > >
> > > [3] One token to fool llm-as-a-judge

---

### Decision · Program_Chairs · 2026-04-30

**Decision:**

Accept (regular)

**Comment:**

The paper studies whether RL-trained language models learn to exploit loopholes in flawed training environments, and reviewers agreed that the topic is timely. The main discussion centered on scope and interpretation: one reviewer remained unconvinced that environments with planted vulnerabilities provide strong evidence about the broader forms of emergent misalignment that matter most in practice, while the other reviewers found the controlled benchmark, together with the rebuttal’s narrowing of claims and clearer positioning relative to real-world RL vulnerabilities, sufficient to make the contribution useful and technically solid. Over the course of discussion, concerns about presentation and overclaiming were largely resolved, the positive reviewers became firmer in their support, although the reviewer with negative score is not fully convinced. Given the average score and the internal discussion, the AC believe it can be accepted.